# Application of Photodynamic Therapy with 5-Aminolevulinic Acid to Extracorporeal Photopheresis in the Treatment of Patients with Chronic Graft-versus-Host Disease: A First-in-Human Study

**DOI:** 10.3390/pharmaceutics13101558

**Published:** 2021-09-26

**Authors:** Eidi Christensen, Olav A. Foss, Petter Quist-Paulsen, Ingrid Staur, Frode Pettersen, Toril Holien, Petras Juzenas, Qian Peng

**Affiliations:** 1Department of Dermatology, St. Olavs Hospital, Trondheim University Hospital, 7030 Trondheim, Norway; Ingrid.Staur@stolav.no; 2Department of Pathology, The Norwegian Radium Hospital, Oslo University Hospital, 0310 Oslo, Norway; Toril.Holien@stolav.no (T.H.); Petras.Juzenas@rr-research.no (P.J.); Qian.Peng@rr-research.no (Q.P.); 3Department of Clinical and Molecular Medicine, Norwegian University of Science and Technology, 7030 Trondheim, Norway; Petter.Quist.Paulsen@stolav.no; 4Department of Orthopaedic Surgery, Clinic of Orthopaedy, Rheumatology and Dermatology, St. Olavs Hospital, Trondheim University Hospital, 7030 Trondheim, Norway; olav.foss@ntnu.no; 5Department of Neuroscience and Movement Science, Norwegian University of Science and Technology, 7030 Trondheim, Norway; 6Department of Haematology, St. Olavs Hospital, Trondheim University Hospital, 7030 Trondheim, Norway; 7Department of Nephrology, St. Olavs Hospital, Trondheim University Hospital, 7030 Trondheim, Norway; Frode.Pettersen@stolav.no; 8Department of Optical Science and Engineering, School of Information Science and Technology, Fudan University, Shanghai 200433, China

**Keywords:** 5-aminolevulinic acid, ALA-based photodynamic therapy, phototherapy, extracorporeal, photopheresis, chronic graft-versus-host disease

## Abstract

Extracorporeal photopheresis (ECP), an immunomodulatory therapy for the treatment of chronic graft-versus-host disease (cGvHD), exposes isolated white blood cells to photoactivatable 8-methoxypsoralen (8-MOP) and UVA light to induce the apoptosis of T-cells and, hence, to modulate immune responses. However, 8-MOP-ECP kills diseased and healthy cells with no selectivity and has limited efficacy in many cases. The use of 5-aminolevulinic acid (ALA) and light (ALA-based photodynamic therapy) may be an alternative, as ex vivo investigations show that ALA-ECP kills T-cells from cGvHD patients more selectively and efficiently than those treated with 8-MOP-ECP. The purpose of this phase I-(II) study was to evaluate the safety and tolerability of ALA-ECP in cGvHD patients. The study included 82 treatments in five patients. One patient was discharged due to the progression of the haematological disease. No significant persistent changes in vital signs or laboratory values were detected. In total, 62 adverse events were reported. Two events were severe, 17 were moderate, and 43 were mild symptoms. None of the adverse events evaluated by the internal safety review committee were considered to be likely related to the study medication. The results indicate that ALA-ECP is safe and is mainly tolerated well by cGvHD patients.

## 1. Introduction

The modification of today’s standard extracorporeal photopheresis (ECP) with the introduction of 5-aminolevulinic acid (ALA) based photodynamic therapy (PDT) may improve treatment efficacy. Since the treatment of cutaneous T-cell lymphoma was approved more than 30 years ago, ECP has been used to treat patients with other T-cell-mediated disorders, including chronic graft-versus-host disease (cGvHD) [1,2,3]. Graft-versus-host disease (GvHD) represents a serious immunologically mediated complication of allogeneic hematopoietic cell transplantation [4]. Donor immune cells, mostly T-lymphocytes (T-cells), are activated and attack the host cells, misrecognising them as foreign. GvHD is classified as chronic based on histological criteria [5]. It can involve multiple organs, of which the skin is the most often affected, and can cause severe morbidity in patients [6]. ECP is widely used in those patients with an initial non or partial response to first-line treatment, including systemic corticosteroids [2].

ECP is a leukapheresis-based therapeutic procedure. The mechanism of its immunomodulatory effect is not fully understood but involves altered T-cell function and the modulation of dendritic cell maturation, with the modification of the cytokine profile and stimulation of regulatory T-cells [2,7,8]. During ECP, leucocytes separated from the whole blood (buffy coat) are exposed to the photosensitising agent 8-methoxypsoralen (8-MOP) and ultraviolet A (UVA) light before reinfusion back into the patient. The combination of 8-MOP-UVA irradiation leads to deoxyribonucleic acid (DNA) crosslinking and, thereby, to the apoptosis of the treated cells. However, a major disadvantage of 8-MOP is that it binds to the DNA of diseased and normal cells with no selectivity, thus inducing the apoptosis of both types of cells after UVA activation. In addition, the treatment is time-consuming, expensive, and often ineffective. This demands the use of an alternative photosensitiser that selectively targets diseased cells to improve the effectiveness of today’s standard treatment.

The idea of using 5-aminolevulinic acid (ALA) as a photosensitiser precursor for ECP has emerged from decades of research and experience with photodiagnosis and PDT. ALA-PDT is well established for topical use in the treatment of patients with non-melanoma skin cancer, and the oral intake of ALA is approved for clinical use in the photodetection of glioma and for the treatment of Barrett’s oesophagus [9,10,11]. ALA is a naturally occurring amino acid and a precursor of heme that is metabolised intracellularly to endogenously porphyrin photosensitisers, mainly protoporphyrin IX (PpIX) [12]. It provides a highly selective accumulation of PpIX in the proliferative or activated cells, including tumour cells, due to their higher uptake of ALA and the alteration of the activities of enzymes involved in heme biosynthesis. The photoactivation of ALA-induced PpIX in the presence of oxygen through the formation of reactive oxygen species leads to the apoptosis and necrosis of the targeted cells [13,14]. After illumination, PpIX is partly or completely photobleached (faded). Under experimental conditions, ALA-UVA is shown to kill T-cells in the blood samples of patients with cGvHD more selectively and efficiently than those treated with 8-MOP-UVA [15]; hence, the number and duration of ECP treatments with the use of ALA can be reduced.

The aim of this study was to evaluate the safety and tolerability of multiple treatments with ALA-ECP in patients with cGvHD. In addition, data are presented from assessments of selected organs, performance, and quality of life in patients during the study period.

## 2. Materials and Methods

### 2.1. Patients

Patients with cGvHD who responded inadequately to 8-MOP-ECP after a minimum of 3 months of treatment at St. Olavs Hospital, Trondheim University Hospital, Norway, were considered for study inclusion. Inadequate response was defined as the progression of cutaneous cGvHD (>25% worsening of skin score from baseline or insufficient response with a <15% improvement in skin score compared with baseline or a ≤25% reduction in corticosteroid dose). Excluded were patients under 18 years of age; with body weight below 40 kg; and those with photosensitive comorbidities such as porphyria or known hypersensitivity to ALA or porphyrins, aphakia, pregnancy or breastfeeding, polyneuropathy, ongoing cardiac and pulmonary diseases, uncontrolled infection or fever, history of heparin-induced thrombocytopenia, an absolute neutrophil count < 1 × 10^9^/L, platelet count < 20 × 10^9^/L, aspartate transaminase (AST), alanine transaminase (ALT), bilirubin, or an international normalised ratio (INR) value ≥ 3× upper limit of normal or significant ECG findings. Patients considered unlikely to comply with the study procedures were also excluded

### 2.2. Design and Procedures

This prospective, open, single-centre, phase I–(II) study was approved by the regional committee for medical research ethics (REK-Nord 2014/2316), the Norwegian Medicines Agency (National Regulatory Authority 14/16760-29) and was registered at www.clinicaltrials.gov (accessed on 20 September 2017) as NCT03109353. The study was performed in accordance with the Helsinki Declaration, monitored by the Clinical Research Unit Central Norway, Norwegian University of Science and Technology (NTNU), and patients provided written informed consent.

The CELLEX Photopheresis machine (Therakos, Mallinckrodt Pharmaceuticals, Raritan, NJ, USA) was used. This is a closed system in which leukapheresis, photoactivation, and the reinfusion of white blood cells are achieved sequentially. The patients were connected to the machine through single or double venous access to collect whole blood. Then, through centrifugation, the fraction of white cells (buffy coat) was separated and collected into a treatment bag (buffy bag), while the other blood components, mainly red cells and plasma, were returned to the patient. A commercially available ALA-hydrochloride powder for systemic oral use (Gliolan, photonamic GmbH & Co. KG, Pinneberg, Germany) was used at a dose of 0.168% w/v (10mM). The pharmacy at St. Olavs Hospital reconstituted ALA-hydrochloride in 0.9% sodium chloride (NaCl) physiological saline under sterile condition to a stock solution of 30 mg/mL. Since the amount of buffy coat collected varied on different days for the same patients and between patients, the volume of the ALA solution added to the buffy coat was adjusted to the volume collected on each treatment to reach a dose of 10 mM. After the ALA-solution was added, a 1-h incubation period in the dark was allowed for intracellular PpIX production before UVA light exposure of the cells. Although the most effective light wavelength for PpIX activation is at 400 nm, the use of the UVA light source in the Therakos photopheresis machine can also effectively kill ALA-incubated human T-cell lines [16].

Finally, the treated buffy coat was reinfused into the patient. One treatment-cycle represents two treatments: one treatment given on two consecutive days. Each patient could receive up to 20 ALA-ECP treatments corresponding to 10 treatment-cycles within a year. The duration between two treatment-cycles varied depending on each patient’s symptoms and treatment response.

Representative buffy coat samples for the investigation of ALA-induced PpIX were taken after a 1-h ALA incubation at room temperature. The plasma samples for PpIX determination were taken immediately after the reinfusion of the ex vivo ALA-ECP-treated buffy coat and 24-h after treatment. Measurements of ALA-induced PpIX in the buffy coat and plasma samples were performed in an extraction solution [17,18]. The solvent consisted of 1% sodium dodecyl sulfate in 1 N perchloric acid (HClO4) and methanol (CH3OH) (1:1 vol/vol). A 100-µL sample was thoroughly mixed with 900 µL of solvent for approximately 1 min in a 1.5-mL tube (Eppendorf, Hamburg, Germany). The sample homogenates were then kept at −20 °C until analysis. Before measurements, the samples were centrifuged for 5 min at 1000 rpm, and the supernatants were carefully transferred to new tubes. The fluorescence of PpIX was measured using a luminescence spectrometer (LS50B, Perkin-Elmer, Norwalk, CT, USA). The PpIX fluorescence in the sample supernatants was measured in a standard 10-mm quartz cuvette placed in the standard holder of the luminescence spectrometer. The fluorescence excitation wavelength was 400 nm, corresponding to the Soret band of the PpIX absorption spectrum. The fluorescence intensity was recorded at 605 nm, corresponding to the PpIX emission peak in the solvent. The amount of PpIX was determined by comparing the PpIX fluorescence intensities with the standard curve made by adding known amounts of PpIX to the solvent. The sensitivity level of our method, that is, the lowest concentration of PpIX that could be detected, was approximately 3 nM in the extraction solution.

### 2.3. Safety and Tolerability

Safety and tolerability were regularly monitored through clinical and laboratory examinations and patient reports (Table 1). Safety was monitored through frequency, seriousness, and intensity of adverse events, 12 lead electrocardiogram (ECG) recordings, vital signs (blood pressure as seated, pulse at rest, and forehead temperature), physical examinations, and laboratory measurements. The schedules of major events within each treatment-cycle and controls are presented in Table 1. Assessments performed either at screening, if it was less than one week before ALA-ECP or performed just before the first treatment were used as reference (baseline). Any abnormality was considered for clinical significance and for the need for action, for example, repetition of the test to verify the result and/or closer follow-up. Laboratory parameters included haematology (haemoglobin, white blood cell (WBC) count, WBC differential count, eosinophil count, platelet count, mean cell volume, mean cell haemoglobin, and INR), and clinical chemistry (albumin, ALT, alkaline phosphatase, AST, bilirubin, blood urea, cholesterol, C-reactive protein (CRP), calcium cation, creatinine, erythrocyte sedimentation rate (SR), gamma-glutamyl transferase, glycated haemoglobin A1c, lactate dehydrogenase, potassium, sodium, and total protein). Urine was examined with a dipstick for erythrocytes, glucose, ketone, leucocytes, nitrite, pH, and protein.

Additional blood tests to analyse AST, ALT, bilirubin, and INR were performed to evaluate the effect of ALA-ECP on the liver. Blood analyses were performed at the Department of Clinical Chemistry, St. Olavs Hospital, and reference values were used. Urine was examined using a dipstick test.

Tolerability was observed through the patients’ reported adverse events (AEs). Any event during the study period in which the patients felt unwell or different from usual was recorded as an AE. Conceivable AEs of systemic ALA treatment included nausea, vomiting, headache, photosensitivity, and chills. Patients received a diary in which AEs were noted between visits. The seriousness of all events was graded (grade 1: asymptomatic or mild symptoms to grade 5; death related to AE) according to Common Terminology Criteria for Adverse Events (CTCAE) v4.03. The assessment of study continuation was based on the stated protocol criteria. An internal safety review committee (ISRC) was appointed to assess safety and tolerability data and to evaluate the reported AEs.

### 2.4. Organ, Performance, and Quality of Life Assessments

Assessments of organ, performance, and quality of life were repeated at approximately 3-month intervals. Investigator-performed evaluations and patient-reported outcome measures were used. The use of immunosuppressive therapy was monitored. Parts of Filipovich and colleagues’ clinical organ scoring system were used in the evaluation of skin manifestations [6]. This scoring system combines clinical features with the affected percentage of the body surface area (BSA) (grade 0: no symptoms; grade 1: <18% BSA with disease signs but no sclerotic features; grade 2: 19–50% BSA or superficial sclerotic features; and grade 3: >50% BSA or deep sclerotic features or impaired mobility, ulceration, or severe pruritus). The affected percentage of BSA was additionally registered in the evaluation of skin disease. Pruritus was assessed using a visual analogue scale (VAS) graded from 1 to 10, with an increasing number corresponding to a higher severity of pruritic manifestation. The diagnostic features of mucosal manifestations were not systematically recorded. Schirmer’s test was used to evaluate tear production. Sterile paper strips were placed into the inferior-temporal aspect of the conjunctival sac of both eyes without topical anaesthesia, and the wetted length was measured in millimetres after 5 min. A measurement of ≥15 mm was considered grade 1 (normal), 14–9 mm as grade 2 (mild), 8–4 as grade 3 (moderate), and less than 4 mm as grade 4 (severe significance). Mouth, gastrointestinal, and performance were assessed using Filipovich’s clinical system for scoring the severity and extent of cGvHD using a 4-point scale (grade 0: no symptoms, grade 1: mild symptoms, grade 3: moderate symptoms, and grade 4: severe symptoms) [6]. Functional status was assessed using Karnovsky’s performance status scale [19]. The patients were given a score on a linear scale between 0 and 100, summarising their ability to perform daily activities: the lower the score was, the worse the status was. Patients were also asked to complete the Skindex-29 and European Organisation for the Research and Treatment of Cancer Quality of Life Questionnaire Core 30 (EORTC30) and the Functional Assessment of Cancer Therapy–Bone Marrow Transplantation (FACT–BMT) questionnaires [20,21,22]. Skindex-29 is designed to measure the overall effect of skin disease on health-related quality of life and is sensitive to minor changes over time. EORTC30 is a widely used measure of cancer-specific health-related quality of life. FACT-BMT is a general measure of cancer-specific health-related quality of life in combination with a module developed specifically for evaluating the quality of life of bone marrow transplant patients.

### 2.5. Statistical Methods

Descriptive data are reported as numbers, percentages (%), and mean (min–max.) and standard deviation (±), together with line graphs and box plots (median, 25–75% percentile). IBM SPSS software version 27 was used.

## 3. Results

### 3.1. Patients and Treatments

Five patients (three women and two men) were included, of which four completed the study. The mean age at entry was 48 years (32–60). Four patients were diagnosed with acute myeloid leukaemia, and one was diagnosed with high-risk myelodysplastic syndrome before undergoing allogeneic bone marrow transplantation. Prior to study inclusion, the patients had received a mean of 25 (10–49) treatment-cycles of 8-MOP-ECP. Three patients received systemic steroids.

During the study period, the patients received a total of 82 single treatments (42 treatment-cycles) with ALA-ECP. One patient missed one treatment due to clotting in the buffy coat. A second patient missed one treatment because of a gastric flu event. The mean volume of the buffy coat collected at treatments was 146 mL (100–251), and the mean irradiation time was 17 min (2–45). As the patients’ symptoms of the disease varied, the treatment intervals were individualised. Two patients had an ECP treatment interval of 4 weeks, each receiving 20 treatments within 9 months. The remaining three patients had an interval of 6–8 weeks and received 18, 13, and 11 treatments, respectively. One patient met a protocol-defined discontinuation criterion 8 months into the study when the treatment with systemic prednisolone was increased from 10/5 mg/every other day to 25 mg/day due to the worsening of symptoms. After discharge from the study, the patient received two 8-MOP-ECP treatments before being diagnosed with lung cancer, after which ECP was discontinued.

### 3.2. Safety and Tolerability

We did not observe any clinically significant changes in the patients’ vital signs. The details of the results are presented in Figure 1.

One patient had an increased QTc interval value of 467 ms on ECG at the 3-month control compared to 450 ms at baseline (prolonged QTc defined as a value > 450 ms [23]). Consequently, an ECG was performed before and after three consecutive treatment cycles, and the patient underwent cardiological examination with an echocardiogram and stress ECG with no pathological findings. Both the cardiologist and ISRC concluded that the QTc interval in this patient was unlikely to be affected by the study’s medicine.

Few clinically significant changes in laboratory values were observed. Two patients had a transient increase in CRP values (34 and 31 mg/L) and SR values (31 and 21 mm/h), respectively, compared with the baseline. One patient had transient high values of eosinophil count (15%), absolute eosinophil count (1.98 × 10^9^/L), and lactate dehydrogenase level (239 U/L). No evidence of liver toxicity was observed (Figure 2).

Most of the bilirubin and INR measurements showed transiently higher values a few days after treatment. High levels of ALT and AST were confirmed at baseline in one patient (patient two) with a history of elevated liver enzymes. This patient regularly used paracetamol but had no established liver disease. Another patient (patient five) with elevated liver enzyme and bilirubin levels four days after treatment had a known bile disorder. No clinically significant long-term liver effects were observed. Albumin and total protein levels were stable in all the patients throughout the study. Dipstick urinalysis revealed no severe kidney or urinary tract disorders. One test yielded a positive urine culture, resulting in antibiotic treatment. ALA-induced PpIX was determined from 75 buffy coat and 104 plasma samples from the five patients and was detected in all the buffy coat samples with a mean concentration of 122 (123) nM. No PpIX was found in the plasma samples taken immediately or at 24 h after re-infusion.

In total, there were 62 cases of AEs (two serious adverse events (SAEs), 20 AEs, and 40 conceivable AEs). Of the 62 events, two (3%) were severe, 17 (27%) were moderate, and 43 (69%) were mild symptoms (Table 2 and Table 3). None of the AEs evaluated by the ISRC were considered to be likely related to the study medication (Table 2). The two SAEs were observed in the same patient. The first event was a pulmonary viral infection occurring 12 days after the second treatment-cycle, and the patient was hospitalised for 14 days. The ISRC evaluated the event as possibly related to the study medication. The second SAE was a migraine headache episode that occurred two days after the sixth treatment-cycle and required two days of hospitalisation. The patient had a history of migraines. The ISRC evaluated the SAE as unlikely to be related to the study medication. The most common grade one and two AEs reported were clotting in the treatment system, cold-like symptoms, and dysuria (Table 2). No grade four or five AEs were reported. The most frequently reported conceivable AEs were nausea and headache (Table 3). Of all the cases, 85% had mild grade one, and 15% had grade two symptoms. No grade 3–5 events were reported.

### 3.3. Organ, Performance, and Quality of Life Assessments

The assessment scores from the baseline and the patients’ last control are shown in Table 4. Most of the conditions showed an improvement in scores with the greatest improvement in skin, BSA involvement, and pruritus. The eye assessment showed the best score at baseline. In one patient, the prednisolone dose of 5 mg/every other day was unchanged, and in another, it was reduced from 10/5 mg/every other day at baseline to 2.5 mg/day at the last control.

## 4. Discussion

This paper reports the results of 82 ALA-ECP treatments administered to five patients with cGvHD who were considered to respond inadequately to 8-MOP-ECP. We did not observe any clinically significant changes in vital signs or detect any persistent abnormal laboratory findings. Overall, we registered an improvement in the patients’ skin scores during the study period.

The rationale for the clinical use of ALA in ECP is multiple. ALA does not induce carcinogenesis, since ALA-induced PpIX localises to the cellular membrane structures outside the nucleus [12,24]. Transformed or activated T-cells produce considerably more ALA-induced PpIX than normal resting cells [15]. As a result, these cells are highly selectively destroyed after light irradiation, while resting normal T-cells remain undamaged. Furthermore, ALA combined with light-based therapy (ALA-based PDT) induces anti-tumour immunity [25]. Besides cell necrosis, it also initiates apoptotic cell death through the mitochondrion and ER-Ca2+-pathways [26], which, in turn, can promote immunosuppressive effects, including regulatory T-cell activation [14]. The highly selective photodamage of transformed or activated T-cells by ALA-PDT with the subsequent immunogenic cell death-mediated induction of an individualised and specific immuno-modulatory response may be a major advantage over non-specific immunosuppressive drugs.

This is the first study of ALA administered to patients treated with ECP. We took the precaution of using a dose of ALA corresponding to 11.2 mg/kg if assuming a 75 kg body weight (10mM). When previously used ex vivo for a 24-h incubation, the same dose gave no dark toxicity to leukocytes from cGvHD patients [16]. Moreover, a single oral dose of 20 mg/kg ALA in drinking water has been approved for the fluorescence-guided resection of glioma [10] in humans, while that of 60 mg/kg has been administered for the PDT treatment of Barrett’s oesophagus [11].

Common side effects observed from the use of oral ALA are nausea, vomiting, and a transient rise in some liver enzymes and bilirubin, which typically resolve after 48 h [27,28]. Similar effects were observed in the present study. The additional finding of a modest increase in the INR value on day two of treatment may be seen in relation to the heparinisation of the patients’ blood in association with ECP. Phototoxicity was not reported as an adverse reaction after treatment and coincides well with the finding of no PpIX in the patients’ plasma.

Conventional ECP with 8-MOP is considered a safe treatment modality with side effects that are commonly sporadic and mild, such as nausea, fever, or headache [29]. Of all the AEs reported in this study, clotting in the buffy coat and episodes of infections were most frequently evaluated to likely be related to the study medicine. We are uncertain whether ALA and/or the 1-h incubation period before the photoactivation of the buffy coat contributed, since abnormal clotting is also recognised with conventional 8-MOP- ECP [30]. Although heparin is part of the standard ECP operational system, the use of anticoagulants varies between hospitals and is adjusted according to the patients’ conditions [31]. Clotting during 8-MOP-ECP is commonly managed at our hospital by increasing the heparin dose. In the four study patients, the heparin dose (5000 IE/mL) was increased from 1.0 to 1.5 mL, then no clotting was observed. No definite association between treatment and the increased chance of infection using ALA-ECP was concluded, although three reported cases of infection were considered to possibly be related. There is no evidence of an increased risk of systemic infection after oral ALA or after conventional ECP [2,30]. The occurrence of secondary malignancies in patients after allogeneic stem cell transplantation has been reported, with an incidence of 5.6% [32]. Second malignancy is not associated with traditional ECP [30,33] and is also not expected using systemic ALA [12,34]. However, unexpected events must be monitored in future larger clinical ALA-ECP studies, the main limitations of the present study being the open-label design and the small patient sample size.

Although safety was the primary focus of this study, organ, performance, and quality of life assessments were regularly performed. It should be emphasised that our observations only reflect changes that occurred during the study period and cannot be attributed to ALA-ECP treatment alone. Factors such as the fluctuating course of the diseases, the comedication of the patients, and the lack of investigator-blinded controls may have influenced the results. Overall, the improvement in scores appears promising, especially for skin. In contrast, tear production was measured to be greatest at baseline. The presence of keratoconjunctivitis sicca is well known in cGvHD [6] and four of the patients used artificial teardrops due to dry eyes before entering the study. Information to patients not to use drops before the baseline visit may have been insufficient. This could explain the finding of lower eye moisture at controls. However, we cannot rule out that the treatment may have affected tear production, and this should be further evaluated in future studies.

This study met the aim to evaluate safety and tolerability after multiple treatments with ALA-ECP in patients with cGvHD and lays the basis for designing future clinical ALA-ECP studies.

In conclusion, the results indicate that ALA-ECP is safe. Treatment appears to be tolerated by patients, with most adverse events reported to be in the mild-to-moderate range of severity. Apart from reduced eye moisture, no findings suggestive of organ toxicity were observed. Further research is needed to assess the safety and for the optimisation of the treatment.

## Figures and Tables

**Figure 1 pharmaceutics-13-01558-f001:**
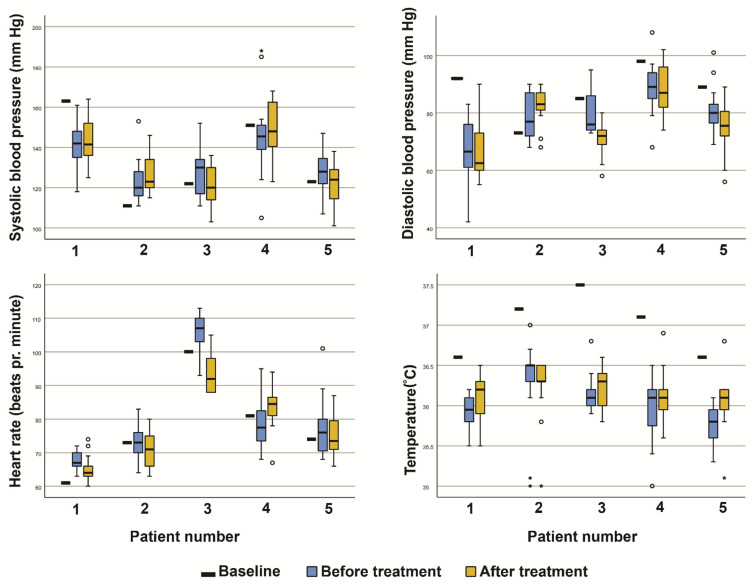
Systolic and diastolic blood pressure, heart rate, and temperature of patients, as measured at baseline and before and after each day in all treatment-cycles. (Boxplot with; median, 25th and 75th percentile, whiskers—value which is not an outlier or an extreme score, ° outlier more than 1.5 box lengths from the box, * extreme case more than 3 box lengths from the box.)

**Figure 2 pharmaceutics-13-01558-f002:**
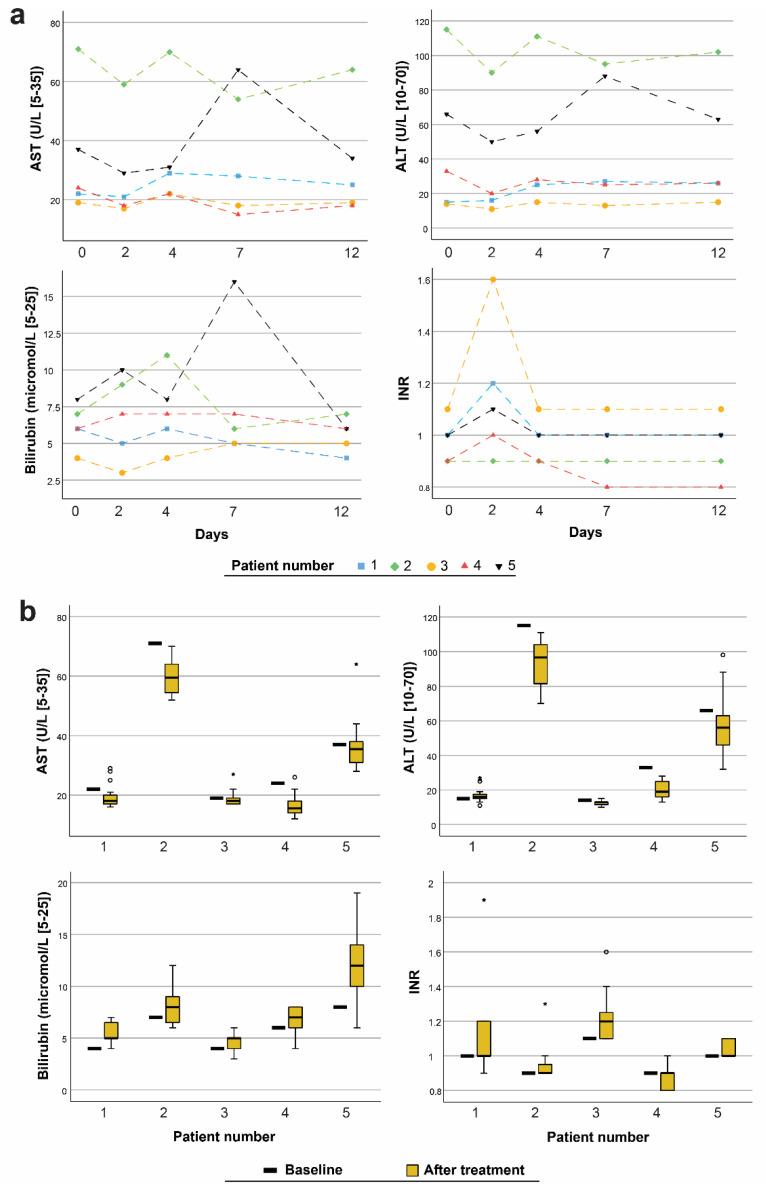
(**a**) Measurements of aspartate transaminase (AST), alkaline phosphatase (ALT), bilirubin, and international normalised ratio (INR) for each patient at baseline and on day 2 of treatment and days 4, 7, and 12 following the first treatment-cycle. (**b**) Measurements of AST, ALT, bilirubin, and INR at baseline and after day 2 of each treatment-cycle. (Boxplot with; median, 25th and 75th percentile, whiskers—value which is not an outlier or an extreme score, ° outlier more than 1.5 box lengths from the box, * extreme case more than 3 box lengths from the box.)

**Table 1 pharmaceutics-13-01558-t001:** Schedule for the assessment of safety, tolerability and response.

	Time Point					
Investigation	Screening (Baseline)	ECP, Day 1	ECP, Day 2	Post-ECP, Day 4 ± 1	Post-ECP, Day 16 ± 2	Controls, Every 3 Months
Vital signs	x	x	x			x
ECG	x					x
Haematology	x	x *				x
Clinical chemistry	x	x *				x
Urine analyses	x	x *				x
Conceivable Adverse Events			x	x		
Adverse Events			x	x	x	x
Organ, performance, and QoL assessments	x	x *				x

* Not needed if less than one week after screening. ECP, extracorporeal photopheresis; ECG, electrocardiogram; QoL, quality of life.

**Table 2 pharmaceutics-13-01558-t002:** Adverse events and Internal Safety Review Committee’s assessments of the severity of the events as well as relation to the study medication.

Type of Adverse Event	Number of Patients	Number of Grade 1	Number of Grade 2	Number of Grade 3	Relation to Study Medication
Clotting in buffy	4	0	4	0	Possibly
Cold like symptoms	3	3	1	0	Unlikely
Dysuria	1	0	3	0	Possible
Respiratory infection	2	0	1	1	Possible
Skin worsening	1	0	1	0	Possible
Migraine	1	0	0	1	Unlikely
Flue like symptoms	1	1	0	0	Unlikely
Elevated INR	1	1	0	0	Unlikely
Prickling around mouth	1	1	0	0	Possible
UVI	1	0	1	0	Possible
Malaise	1	1	0	0	Unlikely
Sore throat, pain toes and hip	1	1	0	0	Unlikely
Prolonged QTc interval	1	1	0	0	Unlikely

Grade 1 severity: mild; asymptomatic or mild symptoms; clinical or diagnostic observation only; intervention not indicated; Grade 2 severity: moderate; minimal, local, or non-invasive intervention indicated; limiting age-appropriate instrumental activities of daily living; Grade 3 severity: severe or medically significant but not immediately life-threatening; hospitalisation or prolongation of hospitalisation indicated disabling, limiting self-care activities of daily living.

**Table 3 pharmaceutics-13-01558-t003:** Frequency and severity of conceivable adverse events among patients.

Type of Adverse Event	Number of Patients	Number of Grade 1	Number of Grade 2
Nausea	4	19	2
Vomiting	2	1	1
Headache	4	10	3
Photosensitivity	0	0	0
Chills	2	4	0

*Nausea* Grade 1: loss of appetite without alteration in eating habits, Grade 2: Oral intake decreased without significant weight loss, dehydration, or malnutrition; *Vomiting* Grade 1: 1–2 episodes in 24 h, Grade 2: 3–5 episodes in 24 h; *Headache* Grade 1: mild pain, Grade 2: moderate pain, limiting instrumental activity of daily living; *Chills* Grade 1: mild sensation of cold, shivering, chatting of teeth.

**Table 4 pharmaceutics-13-01558-t004:** Results of organ, performance, and quality of life assessments at baseline and at the last control for all patients.

Target	Scoring Tool	Scoring Scale	Baseline (Mean Score, min.- max.)	Last Control (Mean Score, min.- max.)
Skin	Modified organ scoring system	0–3 (0 = no symptom)	2.6 (1–3)	1.6 (1–2)
Skin	Body surface area %	0–100% (0 = no area affected)	15.8 (7–30)	5.0 (3–7)
Pruritus	Visual analogue scale	0–10 (0 = no symptom)	3.3 (0–7)	1.4 (1–2)
Mouth	Modified organ scoring system	0–3 (0 = no symptom)	1.6 (0–3)	1.4 (0–2)
Eye, right	Schirmer’s test	1–4 (1 = normal)	3.0 (1–4)	4.0 (4–4)
Eye, left	Schirmer’s test	1–4 (1 = normal)	3.6 (3–4)	4.0 (4–4)
Performance	Modified organ scoring system	0–3 (0 = no symptom)	1.2 (1–2)	0.8 (0–1)
Gastrointestinal tract	Modified organ scoring system	0–3 (0 = no symptom)	0.4 (0–1)	0.4 (0–1)
Function	Karnovsky’s performance scale	1–100 (0 = low function)	74 (70–90)	82 (70–100)
Skindex, emotions	Questionnaire	0–100 (0 = no symptom)	22.0 (8–48)	10.2 (0–30)
Skindex, symptoms	Questionnaire	0–100 (0 = no symptom)	30.4 (21–39)	24.2 (17–36)
Skindex, function	Questionnaire	0–100 (0 = no symptom)	19.6 (2–56)	13.4 (2–42)
Skindex, single item	Questionnaire	0–100 (0 = no symptom)	10.0 (0–25)	10.0 (0–50)
EORTC30, functional	Questionnaire	0–100% (0 = low function)	76.2 (70–100)	82.0 (57–100)
EORTC30, symptoms	Questionnaire	0–100% (0 = low level of symptom)	23.4 (7–40)	20.0 (3–50)
EORTC30, global health status	Questionnaire	0–100% (0 = low state)	65.2 (42–75)	61.4 (16–83)
FACT-G, total score	Questionnaire	0–108 (0 = low state)	86.8 (63–99)	88.4 (70–97)
FACT-BMT, subscale score	Questionnaire	0–40 (0 = much concern)	27.8 (22–35)	29.6 (18–36)

EORTC30: European Organisation for the Research and Treatment of Cancer Quality of Life Questionnaire Core 30, FACT-G: Functional Assessment of Cancer Therapy—Global, FACT-BMT: FACT—Bone Marrow Transplantation.

## Data Availability

No datasets were generated or analysed during the current study.

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
