# Peer review of "Application of Photodynamic Therapy with 5-Aminolevulinic Acid to Extracorporeal Photopheresis in the Treatment of Patients with Chronic Graft-versus-Host Disease: A First-in-Human Study"

_pharmaceutics, 2021, doi:10.3390/pharmaceutics13101558_

Round 1
Reviewer 1 Report
This is a phase I-(II) study to evaluate the safety and tolerability of ALA-ECP in cGvHD patients. The study included 82 treatments in five patients.
Since some of the adverse events were grade 2 or 3, these should also be included in the abstract.
Author Response
Response to reviewer 1:
We thank you for having reviewed our paper, and value the comments made.
The manuscript has been through a special editing service arranged by MDPI.
We have carefully reviewed the results section of the manuscript and hope this has
contributed to make the content more precise and easier to comprehend.
Reporting adverse events is a critical part of conducting a phase I-(II) study. Our study
reports on 62 adverse events of which 69% were of grade 1 (mild), 27% were of grade 2
(moderate) and (3%) were of grade 3 (severe) symptoms. The percentages of grade 1-3
adverse events are now included in the abstract of the revised version of the manuscript.
The general conclusion is drawn based on the results after 82 ALA-ECP treatments showing
no significant changes in patients vital signs og laboratory findings. Tolerability was
observed through the patients’ reported adverse events and from patients diaries in which
AEs were noted between visits. Any event during the study period in which the patients felt
unwell or different from usual was recorded as an AE. With regard to concluding on
tolerance we acknowledge that the term “well-tolerated” is ambiguous. Consequently part of
the concluding text has been reformulated and now state that most of the adverse events are
in the mild-to-moderate range of severity.
Reviewer 2 Report
This manuscript describes the use of 5-ALA as an alternative photosensitizer to 8-methoxypsoralen for extracorporeal photopheresis. Overall, the work is well described and the methodology was well designed. The limited number of patients is the main drawback of this work. However, the main conclusions are supported by the data which showed the safety and tolerability of the proposed treatment. More relevant was the observation of an important improvement of the patient’s skin. Hope that the number of patients would be enlarge in future clinical trials.
Author Response
Response to reviewer 2:
We thank you for having reviewed our paper.
Encouraged by the good results from the present work, we believe it will form the basis for planning of future large clinical studies.
Reviewer 3 Report
This is a completely new and very interesting study. Nevertheless, it is known that PDT induces the inflammatory component, in particular the activity of cyclooxygenase-2. If the authors have the sera of the patients collected during the study, then it is necessary to measure prostaglandin E2 as an inflammatory marker.
Author Response
Response to reviewer 3:
We thank you for having reviewed our paper, and value the comments made.
The manuscript has been through a special editing service arranged by MDPI.
We have carefully reviewed the research design and methods description sections of the manuscript and hope this has contributed to make the content more precise and easier to comprehend.
We recognize that PDT can cause inflammatory response in solid tumours (1, 2). This inflammatory process is complicated with a large number of relevant molecules/mediators and immune cells involved to induce anti-tumour immunity. Prostaglandins (including PGE2), bioactive lipids produced from arachidonic acid by cyclooxygenase enzymes are among those mediators in such inflammatory process. However, ALA-mediated extracorporeal photopheresis (ECP) of buffy coat in the patients with chronic GvHD is an entirely new modality with no data available on the mechanism of action. Our previous study (3) has shown that ALA-PDT selectively and effectively killed activated T cells; while monocyte-derived dendritic cells (DCs) were not affected. Incubation of ALA-PDT damaged T cells with autologous DCs induced a down-regulation of the co-stimulatory molecules CD80/CD86 and also upregulation of IL-10 and indoleamine 2,3-dioxygenase expression, two immunosuppressive factors that may account for the generation of tolerogenic DCs.
In the present manuscript we have focused upon the toxicity and tolerance of ALA-ECP of patients with cGvHD. Parallel to this, an ongoing study has been concentrating on tolerogenic DCs and regulatory T cells (together with relevant mediators and cytokines, etc.), both of which play critical roles in suppressing those wrong/over immune responses of GvHD and many other autoimmune disorders, during and after ALA-ECP. However, this is not the aim of the present study and will be reported later in a separate publication.
References
1. Qian Peng, Johan Moan, Jahn Nesland. Correlation of subcellular and intratumoral photosensitizer localization with ultrastructural features after photodynamic therapy. Ultrastruct Pathol. 1996; 20:109-129.
2. Thomas J. Dougherty, Charles J. Gomer, Barbara W. Henderson, Giulio Jori, David Kessel, Mladen Korbelik, Johan Moan, Qian Peng. Photodynamic Therapy. J Natl Cancer Inst. 1998; 90:889–905.
3. Sagar Darvekar, Petras Juzenas, Morten Oksvold, Andrius Kleinauskas, Toril Holien, Eidi Christensen, Trond Stokke, Mouldy Sioud, Qian Peng. Selective Killing of Activated T Cells by 5- Aminolevulinic Acid Mediated Photodynamic Effect: Potential Improvement of Extracorporeal Photopheresis. Cancers (Basel) 2020; 12:377-298.
Round 2
Reviewer 1 Report
I do not feel that the comments of reviewer #1 about the details of the AEs was adequately addressed in the response.
The abstract minimizes the AEs as reported in the paper in the table.
The abstract overstates that skin conditions improved in line 33, but the focus of the study was on tolerability and safety.
Line 378 and the abstract may overstate that no organ toxicity was noted, since the eyes actually worsened from baseline. This needs to be addressed with speculation on whether the tear ducts are affected by treatment or could be mechanistically and whether this needs to be evaluated further.
Author Response
Response to reviewer 1:
We thank you for having reviewed our paper, and value the comments made.
The manuscript has been through Paperpal and a special editing service arranged by MDPI. With contributions from these services, we hope that the English language has an acceptable standard.
The results section has been improved by clarifying information about the reported adverse events. Below are our answers to the comments that were raised.
Comment: I do not feel that the comments of reviewer #1 about the details of the AEs were adequately addressed in the response.
The abstract minimizes the AEs as reported in the table.
Answer: Information about the reported AEs have been added to the results section of the body manuscript to make the content more clear. The number of EAs of severe, moderate and mild symptoms have been added with reference to tables 2 and 3. Similarly, the text in the abstract on EAs have been changed to correspond with the text in the body manuscript.
Comment: The abstract overstates that the skin conditions improved in line 33, but the focus of the study was on tolerability and safety.
Answer: It is quite correct that the aim of the study was safety and tolerability of ALA-ECP in cGvHD patient. These results are therefore presented in the concluding sentence of the abstract. Information about skin improvement has been deleted from the abstract.
Comment: Line 378 and the abstract may overstate that no organ toxicity was noted, since the eyes actually worsened from baseline. This needs to be addressed with speculation on whether the tear ducts are affected by treatment or could be mechanistically and whether this needs to be evaluated further.
Answer: Four of the patients used artificial teardrops due to dry eyes before entering the study. Information to patients not to use drops before the baseline visit may have been insufficient. Even though this may be a possible explanation for the finding of higher eye moisture at baseline compared to controls, we agree that it cannot be ruled out that this may have been affected by the treatment. Thus, we have deleted the statement of “no organ toxicity”. from the abstract. Also, the finding of reduced eye moisture is now included as a reservation to the conclusion in the body manuscript. Considerations regarding the finding of reduced eye moisture are included to the discussion section of the manuscript.
